# Clinical Relevance of *VEGFA* (rs3025039) +936 C>T Polymorphism in Primary Myelofibrosis: Susceptibility, Clinical Co-Variates, and Outcomes

**DOI:** 10.3390/genes12081271

**Published:** 2021-08-20

**Authors:** Laura Villani, Adriana Carolei, Vittorio Rosti, Margherita Massa, Rita Campanelli, Paolo Catarsi, Carlotta Abbà, Robert Peter Gale, Giovanni Barosi

**Affiliations:** 1Center for the Study of Myelofibrosis, Laboratory of Biochemistry, Biotechnology and Advanced Diagnostics, Istituto di Ricovero e Cura a Carattere Scientifico Policlinico S. Matteo Foundation, 27100 Pavia, Italy; l.villani@smatteo.pv.it (L.V.); a.carolei@smatteo.pv.it (A.C.); v.rosti@smatteo.pv.it (V.R.); r.campanelli@smatteo.pv.it (R.C.); p.catarsi@smatteo.pv.it (P.C.); carlotta.abba@outlook.com (C.A.); 2Laboratory of Biochemistry, Biotechnology and Advanced Diagnostics, Istituto di Ricovero e Cura a Carattere Scientifico Policlinico San Matteo Foundation, 27100 Pavia, Italy; m.massa@smatteo.pv.it; 3Centre for Haematology Research, Department of Immunology and Inflammation, Imperial College London, London SW7 2BU, UK; robertpetergale@gmail.com

**Keywords:** *VEGFA*, rs3025039 polymorphism, primary myelofibrosis, vascular endothelial growth factor, deep vein thrombosis

## Abstract

We evaluated the association of *VEGFA* rs3025039 polymorphism with clinical co-variates and outcomes in 849 subjects with primary myelofibrosis (PMF) and 250 healthy controls. Minor T-allele frequency was higher in subjects with *JAK2*^V617F^ compared with those without *JAK2*^V617F^ (18% vs. 13%; *p* = 0.014). In subjects with *JAK2*^V617F^, the TT genotype was associated at diagnosis with lower platelet concentrations (*p* = 0.033), higher plasma LDH concentration (*p* = 0.005), higher blood CD34-positive cells (*p* = 0.027), lower plasma cholesterol concentration (*p* = 0.046), and higher concentration of high-sensitivity C-reactive protein (*p* = 0.018). These associations were not found in subjects with PMF without *JAK2*^V617F^. In subjects with the TT genotype, risk of death was higher compared with subjects with CC/CT genotypes (HR = 2.12 [1.03, 4.35], *p* = 0.041). Finally, the TT genotype was associated with higher frequency of deep vein thrombosis in typical sites (12.5% vs. 2.5%; OR = 5.46 [1.51, 19.7], *p* = 0.009). In conclusion, in subjects with PMF, the *VEGFA* rs3025039 CT or TT genotypes are more common in those with *JAK2*^V617F^ than in those without *JAK2*^V67F^ mutation and are associated with disease severity, poor prognosis, and risk of deep vein thrombosis.

## 1. Introduction

Primary myelofibrosis (PMF) is a hematological cancer characterized by abnormal proliferation and differentiation of hematopoietic progenitors, variable degrees of bone marrow fibrosis, cytopenias, elevated blood CD34-positive stem and progenitor hematopoietic cells, splenomegaly, and risk of transformation to a blast phase. It is driven by *gain-of-function* mutations in Janus kinase-2 (*JAK2*), calreticulin (*CALR*), or myeloproliferative leukemia virus (*MPL*). These persons sometimes have mutations in genes associated with other hematological cancers including *ASXL1*, *EZH2*, *DMNT3A*, *IDH1* and *IDH2* [1]. However, mutation topography does not completely explain the different phenotypes of PMF, and PMF is associated with different risks of disease progression or transformation.

Angiogenesis is important in the development of cancers and critical in the pathogenesis of PMF [2]. Angiogenesis reflects integrated actions of several vascular growth factors, the most potent being vascular endothelial growth factor A (VEGFA). Considerable data indicate that serum VEGFA concentration is increased in persons with PMF [2,3,4,5,6,7,8,9,10,11]. In humans, *VEGFA* is located on chromosome 6p21.3. Single-nucleotide variants (SNVs) of *VEGFA* affect gene expression by altering key regulatory sequences or by altering mRNA. rs3025039 +936 C>T polymorphism in the 3′-UTR has been studied extensively and is reportedly associated with diverse diseases, including cancers [12,13,14,15,16,17].

It is not known whether there is an association between *VEGFA* rs3025039 genotypes and PMF. We performed a population-based study in 849 subjects with PMF, interrogating correlations between *VEGFA* rs3025039 genotypes and mutation topography, clinical co-variates and outcomes.

## 2. Materials and Methods

### 2.1. Study Population

Stored DNA from blood granulocytes of 849 consecutive subjects with PMF seen at the Center for the Study of Myelofibrosis of the IRCCS Policlinico S. Matteo Foundation in Pavia and included in the institutional database was the primary source material of this study. Clinical data were collected retrospectively on the first visit and prospectively thereafter. In all subjects, the diagnosis was confirmed by a review of the initial bone marrow biopsy. The subjects were classified as PMF based on the WHO diagnostic criteria at the time of their first visit and re-classified according to 2017 revised WHO criteria [18]. After the first examination, visits were scheduled every six months. All subjects gave written informed consent approved by the IRCCS Policlinico S. Matteo Foundation Institutional Ethics Committee for their data to be collected and stored in the database of the Centre for the Study of Myelofibrosis. Controls were healthy Italian subjects belonging to the bone marrow donor transplant registry whose samples were made anonymous for the purpose of the study (*N* = 250).

### 2.2. SNV Analyses

DNA was isolated from blood granulocytes, obtained by density gradient centrifugation, using the QIAamp DNA Blood Mini Kit (QIAGENSciences Inc., Germantown, MD, USA). SNP genotyping was conducted using a predesigned, two-labeled (VIC-FAM) TaqMan Assay C_16198794_10 (Applied Biosystems, Foster City, CA, USA) and the reactions were performed on a CFX96 Real-time PCR Detection System (Biorad Company, Hercules, CA, USA) according to the manufacturer’s instructions.

### 2.3. Data Analyses

Data collected at diagnosis included sex, age, spleen size (cm from the costal margin and spleen index), complete blood count with differential, and serum lactate dehydrogenase and cholesterol concentrations. Blood concentrations of CD34-positive cells, blood CXCR4 expression on CD34-positive cells, and serum concentration of high-sensitivity C-reactive protein (hs-CRP) were also quantified in most subjects [19,20,21].

*JAK2^V^*^617F^ and *MPL*^W515^ were detected by real-time polymerase chain reaction (RT-PCR) or high-resolution melting analyses. *CALR* mutations were identified by capillary electrophoresis and bi-directional sequencing. In selected subjects, we performed cytogenetic testing and tested for mutations in *ASXL1*, *EZH2*, *DMNT3*, *IDH1*, and *IDH2*. Cytogenetic analyses and reporting were performed according to the International System for Human Cytogenetic Nomenclature criteria using standard techniques [22]. Next-generation sequencing (NGS) was performed using the diagnostic panel commercially available Oncomine Myeloid Research Assay^®^ (Thermo Fisher, Waltham, MA, USA). Genomic and transcript analyses were performed with IonReporter^®^ software by applying the latest release of the Myeloid workflow (Thermo Fisher, Waltham, MA, USA). Variations causing missense, frameshift, an altered stop/initiation codon, in-frame insertion/deletion or variants affecting splice sites were regarded as mutations. In all subjects, bone marrow biopsies were analyzed at diagnosis and before any therapy started. Grading of bone marrow fibrosis was determined according to the EUMNET consensus [23].

### 2.4. Statistical Analyses

Continuous variables were presented as median with interquartile range (IQR). Pearson’s χ test with one degree of freedom was employed to compare allele and genotype frequencies between cases and controls. Deviations from the Hardy–Weinberg equilibrium within cases and controls were tested by Fisher’s exact test. Dominant models (presence vs. absence of the minor allele), and recessive models (presence vs. absence of two copies of the minor allele) were used. The independent contribution of rs3025039 SNV to odds of PMF phenotypes was assessed by logistic regression analysis. Because this is an exploratory study, we did not introduce a correction for multiple comparisons [24].

To test whether the *VEGFA rs3025039* genotypes correlated with hematological co-variates, we analyzed the risk of developing a hemoglobin concentration <100 g/L, a platelet concentration <150 × 10^9^/L, a WBC <4 × 10^9^/L or >12 × 10^9^/L, blood CD34-positive cells >100 × 10^6^/L or splenomegaly >10 cm below left costal margin. The subjects were censored at the time of therapy or last follow-up visit.

We also analyzed the occurrence of major thromboses including arterial thromboses (myocardial infarction and stroke), venous thromboses (deep vein thromboses and pulmonary embolism), and thromboses in atypical sites (portal vein thrombosis, Budd–Chiari syndrome, and cerebral sinus thrombosis). Survival was calculated as the time from diagnosis until death. A log-rank test was used to test for survival differences for subjects with different *VEGFA* rs3025039 genotypes. The hazard ratio with 95% confidence interval (CI) for different genotypes was estimated by Cox regression analysis. Computations were performed with STATISTICA© software (Dell Technologies Inc., Round Rock, TX, USA).

## 3. Results

Table 1 displays demographic and clinical co-variates of the study subjects. In all, 503 subjects were male (59%) with a median age of 52 years (IQR, 46–61 years), and 484 subjects (57%) had *prefibrotic* myelofibrosis. The IPSS risk distribution was 61% low, 17% intermediate-1, 14% intermediate-2, and 8% high [25].

### 3.1. Correlation between VEGFA rs3025039 Genotypes and PMF

*VEGFA* rs3025039 genotypes in subjects with PMF and normals were consistent with the Hardy–Weinberg equilibrium (*p* = 0.86 and *p* = 0.54). Neither the frequencies of *VEGFA* rs3025039 genotypes nor those of the minor T-allele of the SNV differed significantly between the PMF subjects and local healthy controls (Table 2).

### 3.2. VEGFA rs3025039 Genotypes and Somatic Driver Mutations

PMF cohorts defined by the somatic driver mutations including *JAK2*^V617F^*, CALR, MPL* and the so-called *triple negative* genotypes were in Hardy–Weinberg equilibrium (*p* > 0.05). In subjects with *JAK2*^V617F^ the frequency of the *VEGFA* rs3025039 minor T-allele was like that of normals and of the entire PMF cohort (Appendix A Appendix A). However, T-allele frequency was higher in subjects with *JAK2*^V617F^ compared with those without *JAK2*^V617F^ (18% vs. 13%; OR = 1.43 [1.07, 1.91], *p* = 0.014). In addition, in subjects with *JAK2*^V617F^, the frequency of CT and TT genotypes was higher compared with that of the CC genotype (73% vs. 63%; OR = 1.58, 95% [1.41, 2.20], *p* = 0.006). Adjusting for confounding by age and sex did not alter the findings. In subjects without *JAK2*^V617F^, there was a lower frequency of the T-allele genotype compared with the normals (13% vs. 18%; OR = 0.68 [0.48, 0.94], *p* = 0.02).

### 3.3. VEGFA rs3025039 Genotypes and Clinical Co-Variates at Diagnosis

Sex, age, hemoglobin concentration, and WBC distributions were not significantly correlated with *VEGFA* rs3025039 genotypes (Appendix A). Consequently, no association was detected between the polymorphism genotypes and IPSS prognostic score. However, the subjects with a TT genotype had a lower platelet concentration (median, 281 vs. 474 × 10^9^/L; *p* = 0.02) and a higher plasma LDH concentration (1.58 vs. 1.26 times ULN; *p* = 0.04). In total, 33% of the subjects with the TT genotype had platelet concentration <150 × 10^9^/L, whereas in the subjects with the CC/CT genotypes, this proportion was 14% (OR = 3.10 [1.29, 7.42], *p* = 0.01). The subjects with a TT genotype had an increased plasma LDH concentration compared with 66% in the subjects with the CC/CT genotypes (*p* = 0.076). The subjects with the CT/TT genotypes had more frequent bone marrow fibrosis grade 2–3 compared with the subjects with the CC genotype (48% vs. 41; OR, 1.34 [1, 1.81], *p* = 0.05).

We tested whether these associations were influenced by driver mutation. In the subjects with *JAK2*^V617F^ these associations persisted (Appendix A). In addition, the subjects with *JAK2*^V617F^ and the TT genotype had higher blood concentration of CD34-positive cells compared with other genotypes (median, 28 vs. 7 × 10^6^; *p* = 0.03). Moreover, the subjects with CT/TT genotypes had lower plasma cholesterol concentration (median, 146 vs. 160 mg/dL; *p* = 0.05) and higher plasma hs-CRP concentration compared with the subjects with the CC genotype (median, 0.31 vs. 0.12; *p* = 0.02). None of these associations were detected in the subjects with driver mutations other than *JAK2*^V617F^ even when the sample size of the comparisons provides sufficient statistical power (Appendix A).

### 3.4. VEGFA rs3025039 Genotypes and Outcomes

After a median follow-up of 77 months (IQR, 36–150 months), 364 subjects (43%) developed a hemoglobin concentration <100 g/L, 42% splenomegaly >10 cm below the left costal margin, 41%, a WBC > 12 × 10^9^/L, 30% platelet concentration <150 × 10^9^/L, 15%, a WBC < 4 × 10^9^/L, and 34%, blood CD34-positive cells >100 × 10^6^/L. A total of 102 subjects (12%) had a splenectomy, 93 (11%) received a hematopoietic cell transplant, 155 (18%) had blast transformation, and 226 (27%) died. Median survival was 21 years (IQR, 10, 32 years).

Hazards of developing a hemoglobin concentration <100 g/L, splenomegaly >10 cm from the costal margin, WBC > 12 × 10^9^/L, or WBC < 4 × 10^9^/L, did not differ between *VEGFA* rs3025039 genotypes. However, T-allele frequency stratified the subjects into two well-delineated risk groups for the development of thrombocytopenia (recessive model, HR = 2.17 [1.25, 3.85], *p* = 0.0064; Table 3). In total, 11 of 24 subjects with the TT genotype had a platelet concentration <150 × 10^9^/L at diagnosis or developed it after diagnosis compared with 238 of 821 subjects with CC/CT genotypes. Association of the TT genotype with risk of thrombocytopenia was significant only in the subjects with *JAK2*^V617F^ (HR = 2.5 [1.35, 5.0], *p* = 0.004; Appendix A). In addition, in these subjects, the CT/TT genotype was associated with the risk of developing a spleen >10 cm below the costal margin (HR = 1.31 [1.01, 1.72], *p* = 0.05).

Risk of death was higher in the subjects with the TT genotype compared to the subjects with the CC or CT genotype (OR = 1.92; [1.06, 3.45], *p* = 0.03; Table 3; Figure 1). However, this increased risk was evident only in the subjects with *JAK2*^V617F^ (Appendix A). The few events in the subjects without *JAK2*^V617F^ precluded critical testing.

### 3.5. VEGFA rs3025039 Genotypes and Thromboses

In total, 170 subjects (20%) had a major thrombotic event (Table 4), 97 (57%) of which were vein thrombosis in atypical sites, 49 (29%) arterial thrombosis, and 24 (14%) vein thrombosis in typical sites. Incidences did not correlate with *VEGFA* rs3025039 genotype. However, a higher frequency of deep vein thrombosis in typical sites was detected in the subjects with the TT genotype compared with other genotypes (12.5% vs. 2.5%; OR = 5.46 [1.51, 19.7], *p* = 0.0096).

## 4. Discussion

We report correlations between *VEGFA* rs3025039 genotypes with disease frequency, clinical co-variates at diagnosis, and outcomes in subjects with PMF. According to the WHO classification, the PMF cohort includes prefibrotic myelofibrosis and overt myelofibrosis. We show that PMF subjects with the *rs3025039* minor T-allele have an increased susceptibility to the *JAK2*^V617F^ driver mutation, in keeping with data indicating that other genetic polymorphisms in *JAK2*, *MECOM*, *TERT*, *TET2*, *HBS1L-MYB*, and the corticosteroid receptor predispose to acquiring *JAK2*^V617F^ [26,27,28,29,30,31,32,33].

In the subjects with *JAK2*^V617F^, the TT genotype was associated with more severe disease at diagnosis, i.e., lower platelet concentration, higher plasma LDH concentration, splenomegaly, increased blood concentration of CD34-positive cells, lower serum cholesterol concentration, more inflammation evidenced by higher plasma hs-CRP concentration, and an increased risk of developing thrombocytopenia. These factors translated to worse survival in subjects with *JAK2*^V617F^.

These data suggest a functional role for *VEGFA* rs3025039 SNV in PMF severity and progression. They also suggest that this influence operates predominantly in persons with *JAK2*^V617F^ but not in other driver mutations. However, this observation needs confirmation because of the relatively few subjects with *MPL* mutations and those who were *triple negative*.

How *VEGFA* expression affects PMF disease activity and progression is unknown. The minor T-allele of the *VEGFA* rs3025039 in the 3′-UTR is associated with lower plasma VEGFA concentrations in normals and disease [34,35,36]. Other data suggest a discordance between *VEGFA* expression and VEGFA plasma and that the other genes involved in VEGFA signaling are also important in cancer susceptibility and phenotype [37,38,39].

Pathways active by *JAK2*^V617F^ and *CALR* mutation in PMF are similar but not identical. For example, *CALR* mutations interact with the thrombopoietin receptor (MPL), a protein involved in calcium regulation and processing of IL-6 receptors, interactions not found in *JAK2*^V617F^ [40,41]. In addition, different mechanisms of evasion of immune surveillance are reported [42,43,44,45]. We recently reported that myeloid expansion, blast transformation, and survival correlate with sIL-2Rα-mediated immune escape in subjects with *JAK2*^V617F^ but not those with a *CALR* mutation [46].

The subjects with the *VEGFA* rs3025039 TT genotype had a higher risk of deep vein thrombosis in typical sites. Current knowledge about the potential impact of the rs3025039 SNV in *VEGFA* on angiogenesis and endothelial function is still very limited; thus, the explanation for the observed relationship with thrombosis remains highly speculative.

Our study has limitations. It is retrospective and predominantly descriptive, limiting the certainty of some of our conclusions. Moreover, the numbers of subjects with *MPL* mutations or those who were *triple negative* were small, limiting our power to detect some effects. Further studies are needed to explore the effects of *VEGFA* genotypes and gene–gene interactions in PMF and to validate our findings [47].

## 5. Conclusions

In conclusion, we found that persons with PMF and *VEGFA* rs3025039 minor T-allele genotypes are more likely to have a *JAK2*^V617F^ compared to other driver mutations, and different clinical co-variates at diagnosis and clinical outcomes. These findings help to enrich our knowledge about the genetic basis of PMF and, if validated, they have relevant clinical and prognostic implications.

## Figures and Tables

**Figure 1 genes-12-01271-f001:**
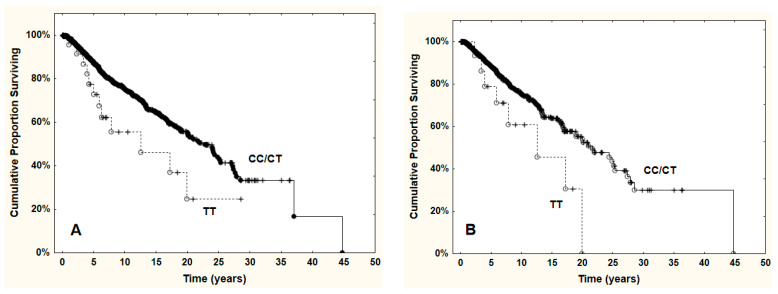
(**A**) Probability of survival in subjects with PMF stratified for the *VEGFA rs3025039* polymorphism genotypes. (**B**) Probability of survival in subjects with PMF bearing the *JAK2*^V617F^ mutation, stratified for the *VEGFA rs3025039* polymorphism genotypes. The subjects with the TT genotype had a shorter survival than those with the CC or CT genotype (*p* = 0.029 and *p* = 0.036, respectively).

**Table 1 genes-12-01271-t001:** Baseline co-variates (*N* = 849).

Demographic Co-Variates	
Age, y, median (IQR)	52 (46–61)
Male, *N* (%)	503 (59)
**Laboratory Co-Variates**	
Hemoglobin, g/L, median (IQR)	131 (109–148)
WBC × 10^9^/L, median (IQR)	8.5 (6.4–11.6)
Platelets × 10^9^/L, median (IQR)	467 (246–711)
Monocytes × 10^9^/L, median (IQR)	496 (332–688)
Spleen size, cm^2^, median (IQR) ^a^	120 (90–160)
Plasma LDH, × upper limit of normal (ULN), median (IQR) ^b^	1.29 (0.92–1.95)
Serum cholesterol, mg/dL, median (IQR) ^c^	158 (130–183)
Plasma high-sensitivity C-reactive protein, mg/dL, median (IQR^) d^	10 (4–44)
Blood CD34-positive cells, × 10^6^/L, median (IQR) ^e^	10 (4–43)
CXCR4/CD34, %, median (IQR) ^f^	41(21–63)
**Molecular Co-Variates**	
*JAK2*^V617F^-positive, *N* (%)	544 (66)
*JAK2*^V617F^ allele frequency, median (IQR)	41 (22–68)
*CALR* mutation, *N* (%)	171 (21)
*MPL* mutation, *N* (%)	44 (5)
*Triple negative*, *N* (%)	68 (8)
NGS detected mutations, *N* (%) ^g^	46 (20)
Cytogenetic abnormalities, *N* (%) ^h^	87 (30)
**Bone Marrow Fibrosis Grade**	
0, *N* (%)	258 (30)
1, *N* (%)	226 (27)
2, *N* (%)	245 (29)
3, *N* (%)	117 (14)

^a^ Spleen size was measured using the spleen index calculated by multiplying the length of the longitudinal axis by the transverse axis. ^b^ Plasma lactic dehydrogenase activity (LDH) was available in 474 subjects. ^c^ Serum cholesterol concentration was available in 422 subjects. ^d^ Plasma high-sensitivity C-reactive protein was available in 237 subjects. ^e^ Blood CD34-positive cell concentration was available in 402 subjects. ^f^ CXCR4/CD34 ratio was available in 298 subjects. ^g^ Next-generation sequencing (NGS) data were available in 242 subjects. ^h^ Cytogenetics was available in 291 subjects.

**Table 2 genes-12-01271-t002:** *VEGFA* rs3025039 genotypes.

		CC	CT	TT	CC/CT	CT/TT	T-Allele Frequency
PMF, *N* (%)	849	601 (70.8)	224 (26.4)	24 (2.8)	825 (97.2)	248 (29.2)	272/1698 (16)
Controls, *N* (%)	250	165 (66)	79 (31.6)	6 (2.4)	244 (97.6)	85 (34)	91/500 (18.2)

**Table 3 genes-12-01271-t003:** Hazard ratio of the outcomes of subjects with PMF stratified according to the *VEGFA* 3025039 genotype.

	CT/TT (*N* = 248) vs. CC (*N* = 601)		TT (*N* = 24) vs. CC/CT (*N* = 825)	
Outcome	HR (95% CI)	*p*-Value	HR (95% CI)	*p*-Value
Hemoglobin < 100 g/L	1.16 (0.92, 1.45)	0.20	1.41 (0.80, 2.44)	0.23
Spleen > 10 cm below left costal margin	1.22 (0.97, 1.51)	0.10	1.01 (0.51, 1.96)	0.99
WBC > 12 × 10^9^/L	1.10 (0.87, 1.37)	0.45	1.02 (0.52, 1.98)	0.94
WBC < 4 × 10^9^/L	1.07 (0.73, 1.58)	0.72	1.26 (0.46, 3.45)	0.65
Platelets < 150 × 10^9^/L	1.12 (0.93, 1.51)	0.39	2,17 (1.25, 3.85)	0.006
Blood CD34-positive cells > 100 × 10^6^/L	1.25 (0.93, 1.51)	0.11	1.11 (0.46, 2.70)	0.81
Transplant	1.49 (0.98, 2.27)	0.07	1.39 (0.44, 4.35)	0.58
Blast transformation	1.15 (0.80, 1.65)	0.43	1.22 (0.49, 2.94)	0.67
Death	1.31 (1.00, 1.72)	0.12	1.92 (1.06, 3.45)	0.03

**Table 4 genes-12-01271-t004:** Major thrombotic events stratified by the *VEGFA* rs3025039 genotype.

		*VEGFA* rs3025039 Genotype					CT/TT vs. CC OR (95% CI)	TT vs. CC/CT OR (95% CI)
	All Subjects (*N* = 848)	CC (*N* = 600)	CT (*N* = 224)	TT (*N* = 24)	CC/CT (*N* = 824)	CT/TT (N = 248)		
Thrombotic events, *N* (%)	170 (20)	124 (20.7)	41 (18.3)	5 (20.8)	165 (20)	46 (18.5)	OR = 0.87 (0.60, 1.27) *p* = 0.48	OR = 1.05 (0.38, 2.85) *p* = 0.92
Arterial thrombosis, *N* (% of PMF cases)	49 (5.8)	36 (6)	12 (5.3)	1 (4.2)	48 (5.8)	13 (5.2)	OR = 0.87 (0.45, 1.66) *p* = 0.67	OR = 0.70 (0.09, 5.31) *p* = 0.73
- In the year before diagnosis, *N* (% of thromboses)	14 (28.6)	10 (27.7)	4 (33.3)	0 (0)	14 (29.2)	4 (30.7)		
- At diagnosis, *N* (% of thromboses)	12 (24.5)	9 (25)	2 (16.7)	1 (100)	11 (22-9)	3 (23.1)		
- After diagnosis, *N* (% of thromboses)	23 (46.9)	17 (47.2)	6 (50)	0 (0)	23 (48)	6 (46.1)		
Deep vein thrombosis in typical sites, *N* (% of PMF cases)	24 (2.8)	17 (2.8)	4 (1.8)	3 (12.5)	21 (2.5)	7 (2.8)	OR = 0.99 (0.41, 2.43) *p* = 0.99	OR = 5.46 (1.51, 19.7) *p* = 0.0096
- In the year before diagnosis, *N* (% of thromboses)	3 (12.5)	1 (5.9)	1 (25)	1 (33.3)	2 (9.5)	2 (28.6)		
- At diagnosis, *N* (% of thromboses)	5 (28.8)	5 (29.4)	0 (0)	0 (0)	5 (23.8)	0 (0)		
- After diagnosis, *N* (% of thromboses)	16 (66.6)	11 (64.7)	3 (75)	2 (66.6)	14 (66.6)	5 (71.4)		
Vein thrombosis in atypical sites, *N* (% of PMF cases)	97 (11.4)	71 (11.8)	25 (11.1)	1 (4.2)	96 (11.6)	26 (9.7)	OR = 0.87 (0.54, 1.40) *p* = 0.57	OR = 0.33 (0.04, 2.47) *p* = 0.28
- In the year before diagnosis, *N* (% of thromboses)	8 (8.2)	6 (8.5)	2 (8)	0 (0)	8 (8.3)	2 (7.7)		
- At diagnosis, *N* (% of thromboses)	73 (75.2)	53 (74.7)	19 (76)	1 (100)	72 (75)	20 (76.9)		
- After diagnosis, *N* (% of thromboses)	16 (16.5)	12 (16.9)	4 (16)	0 (0)	16 (16.7)	4 (15.4)		

PMF = primary myelofibrosis; OR = odds ratio.

## Data Availability

The data presented in this study are available on request from the corresponding author.

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
