# Peer review of "Clinical Relevance of VEGFA (rs3025039) +936 C>T Polymorphism in Primary Myelofibrosis: Susceptibility, Clinical Co-Variates, and Outcomes"

_genes, 2021, doi:10.3390/genes12081271_

Round 1

Reviewer 1 Report

This manuscript evaluated the association of VEGFA rs3025039 polymorphism with clinical co-variates and outcomes in 849 people with PMF (including prefibrotic MF) and 250 healthy controls. They found that the TT or CT genotypes are more common in those with the JAK2V617F mutation and are associated with disease severity, poor prognosis, and risk of deep vein thrombosis.

  1. Why was there a focus on a single polymorphism in VEGFA rather than investigating a number of polymorphisms in this gene? Also, why was the focus not expanded to other genes in the VEGF signaling pathway?
  2. The fact that > 50% of the subjects had prefibrotic MF rather than overt MF is buried and may be missed by a reader. It would be important to make that point more clear throughout the paper.
  3. Why were post-PV and post-ET MF not included? Also, why were PV and ET not included? It would be interesting to compare subjects with longstanding PV or ET without progression to MF vs those who have transitioned to MF to determine if VEGFA polymorphisms may be more common among PV/ET who have transitioned to MF. Also, the correlation of the VEGFA rs3025039 on thrombosis in PV and ET would be interesting to investigate. I understand these analyses may be beyond the scope of this current paper.
  4. NGS sequencing is noted to have been performed in a subset of patients in materials and methods as well as in Table 1, however the only information regarding this is in the form of % of patients with mutations. Was any analysis performed to determine if specific gene mutations (for example high risk genes) were more common among those with a specific rs3025039 polymorphism?

Reviewer 2 Report

This is a well written and straightforward assessment of the clinical relevance of VEGFA polymorphisms in primary myelofibrosis. The authors provide clinical characteristics associated with these polymorphisms. 

Questions/comments as follows:

Major:

  1. The authors provide a number of clinical variables to compare and contrast the presence of this VEGFA polymorphism. The most notable clinical scoring system in primary myelofibrosis is the DIPSS plus which provides a dynamic scoring system that can be applied at different time points in the patient's care. The alternative would be to use the DIPSS scoring system. I would be interested to know how patients with TT fared when this was taken into consideration.
  2. Why do the authors think patients with a TT genotype would be more likely to have venous thrombosis at typical sites? Is there data to support an increased risk for VTE at typical and not atypical or arterial events in patients with MPNs or do the authors have an alternative hypothesis?
  3. How many patients were included in the study? Tables 2 and 3 say 849 with 601 patients with CC genotype while Table 4 has 848 patients with 600 patients with CC. This seems to be an error perhaps due to incomplete information on one patient. 

Minor

  1. Instead of blast transformation I would favor transformation to Acute Myeloid Leukemia.
  2. The X axis of the survival graph would be better displayed as either years rather than months or in 12 month blocks.
  3. For Figure 1 I would include 2 KM curves with one for all patients and another for JAK2 mutation to more clearly show the effect of TT genotype on survival in this population. 
